# Identifying the outbreak signal of COVID-19 before the response of the traditional disease monitoring system

Yaoyao Dai[1,2], Jianming Wang[2]*

1 Department of Infectious Diseases, Center for Disease Control and Prevention of Nantong City, Nantong, China, 2 Department of Epidemiology, Center for Global Health, School of Public Health, Nanjing Medical University, Nanjing, China

* jmwang@njmu.edu.cn

**Data Availability Statement:** All relevant data are within the manuscript and its Supporting Information files.

**Funding:** This study was funded by the National Natural Science Foundation of China (81973103),

## Abstract

New coronavirus cases and related deaths are continuing to occur worldwide. Early identification of the emergence of novel outbreaks of infectious diseases is critical to the generation of timely responses. We performed a comparative study to determine the feasibility of the early detection of the COVID-19 outbreak in China based on influenza surveillance data and the internet-based Baidu search index to evaluate the timelines of the alert signals compared with the traditional case reporting and response systems. An abnormal increase in the number of influenza-like illnesses (ILI) occurred at least one month earlier than the clinical reports of pneumonia with unknown causes and the conventional monitoring system. The peak of the search volume was 20 days earlier than the issuance of the massive official warning about the epidemic. The findings from this study suggest that monitoring abnormal surges of ILI and identifying peaks of online searches of key terms can provide early signals of novel disease outbreaks. We emphasize the importance of broadening the potential of syndromic surveillance, internet searches, and social media data together with the traditional disease surveillance system to enhance early detection and understanding of emerging infectious diseases.

## Synopsis

Early identification of the emergence of an outbreak of a novel infectious disease is critical to generating a timely response. The traditional monitoring system is adequate for detecting the outbreak of common diseases; however, it is insufficient for the discovery of novel infectious diseases. In this study, we used COVID-19 as an example to compare the delay time of different tools for identifying disease outbreaks. The results showed that both the abnormal spike in influenza-like illnesses and the peak of online searches of key terms could provide early signals. We emphasize the importance of testing these findings and discussing the broader potential to use syndromic surveillance, internet searches, and social media data together with traditional disease surveillance systems for early detection and understanding of novel emerging infectious diseases.

National Key R&D Program of China
(2017YFC0907000), Key Project of Philosophy and
Social Science Research in Colleges and
Universities in Jiangsu Province (2020SJZDA096),
and Priority Academic Program Development of
Jiangsu Higher Education Institutions (PAPD). The
funders had no role in study design, data collection
and analysis, decision to publish, or preparation of
the manuscript.

**Competing interests:** The authors have declared
that no competing interests exist.

## Introduction

New coronavirus cases and related deaths are continuing occur worldwide.[1] The WHO, on March 11, 2020, declared the coronavirus disease 2019 (COVID-19) outbreak a global pandemic. This pandemic dates back to December 2019, when a cluster of unexplained pneumonia cases was identified, which were linked to a seafood market in Wuhan, China.[2] Subsequent investigations determined that a novel coronavirus, severe acute respiratory syndrome coronavirus 2 (SARS-CoV-2), was the causative agent now at the heart of the pandemic of an emerging infectious disease (EID). The virus jumped from the transportation hub to other areas during the peak seasonal travel periods of the winter holiday and the traditional Spring Festival.[3] To control the spread and mitigate the risk of the virus, a series of strong, unprecedented measures were taken by the Chinese government. These measures included the mandatory wearing of face masks in public, canceling of mass events, closing of scenic attractions, suspending of long-distance buses, and asking hundreds of millions of Chinese citizens to stay indoors to break the transmission chain.[4, 5] Despite the rapid increase in the number of COVID-19 cases in January, China has now passed the peak of the epidemic and has effectively controlled the disease.[4] No new infections of the novel coronavirus were reported on March 18 in Wuhan, the epicenter of the epidemic in China, marking a notable first success in the months-long battle with the virus and showing hope of suppressing the pandemic.

Because this is an infectious disease caused by a new virus, it took approximately one month from the initial detection of unexplained pneumonia cases to the definite conclusion of "human-to-human transmission" and the inclusion of the disease in the management of statutory infectious diseases by the National Health Commission, China.

The traditional disease monitoring system is useful for detecting the outbreak of common infectious diseases, but it is insufficient for the discovery of new diseases.[6] How to build a comprehensive early warning system of public health emergencies from multiple sources has become the focus of attention of all countries.

To compensate for the shortcomings of the traditional disease monitoring system, some scholars have tried to use digital data streams, [7] network density, [8] and Google Trends (GT) [9] as early warning indicators; these attempts have achieved remarkable results; nevertheless, the roles of these indicators in COVID-19 remain unclear. In this study, we performed a comparative study to discuss the **early** warning capability, timelines, and validity of alert **signals** for the first wave of the COVID-19 outbreak in China based on the surveillance data of influenza-like illness (ILI) and the Baidu Search Index (BSI) compared with the traditional case reporting system.

## Methods

### The data source of COVID-19

COVID-19 data from China were obtained from the Center for Disease Control and Prevention of China and National Health Commission of China as well as the Report of the WHO-China Joint Mission on Coronavirus Disease 2019 (https://www.who.int) and Vital Surveillances Report on the Epidemiological Characteristics of an Outbreak of COVID-19—China, 2020.[10]

### The data source of influenza-like illnesses

We extracted data regarding ILI reported from January 2015 to May 2019 from the National Health Commission of China. After the 2003 SARS epidemic, the Chinese government built the world's most extensive internet-based disease reporting system, called the China

Information System for Disease Control and Prevention (CISDCP).[11] Cases of infectious diseases, categorized as class A, B, and C, are required to be reported through the CISDCP within a limited time. We compared the monthly morbidity of ILI during the last five years and plotted a line chart to describe the long-term trend. We also compared the peak of ILI with the onset of the COVID-19 in the late 2019 in China.

### The data source of the internet-based search index

We used the Baidu search engine (http://index.baidu.com/v2/#/) to analyze the BSI for searches of the keywords of "pneumonia" and "SARS" from November 1, 2019, to February 1, 2020. Baidu is the world's largest Chinese search engine and China's largest internet integrated service company. The BSI reflects active searches by internet users. We compared the timeline of peak searches for these key terms with the time of official response to the epidemic.

### Statistics

Data were entered into Excel and analyzed using SPSS 25 (IBM, NY, USA). The ILI cases across several years were compared using the analysis of variance. The Dunnet method was used for pairwise comparison. The test level for significance was set at 0.05.

### Ethical approval

Data of this study were extracted from a public database. No individual information was published in this paper. Therefore, this study is exempt from ethical approval.

## Results

### The response of the traditional public health emergency reporting system to the outbreak of COVID-19

On December 29, 2019, the Department of Health of Hubei Province and Wuhan city received a report from a local hospital regarding patients with unexplained pneumonia, all of whom were employees of the South China seafood wholesale market. On December 31, the National Health Commission and CDC sent a team of experts to Wuhan. The investigators excluded several suspected causes, including influenza, avian influenza, adenovirus, severe acute respiratory syndrome coronavirus (SARS-CoV), and the middle east respiratory syndrome coronavirus (MERS-CoV). On January 1, 2020, the local government closed this seafood market and disinfected the area. On January 3, 2020, the Chinese government informed the WHO of the outbreak of unexplained pneumonia. On January 7, 2020, the pathogen was identified as a new type of coronavirus, and then, the full genome sequences of this new virus were shared. On January 10, an expert group and a WHO team were invited to visit Wuhan for a field investigation. By January 19, 198 novel coronavirus cases have been reported in Wuhan. As of January 19, the risk of human-to-human transmission of this new virus had not been determined, and officials have not realized the potential global epidemic risk. On January 20, the novel coronavirus pneumonia was incorporated as a notifiable disease under the Infectious Disease Law and Health and Quarantine Law in China. On January 23, the whole city of Wuhan was locked down, and all the residents were required to stay at home. Two days later, the Chinese government made the highest-level commitment to mobilize all forces to stop the epidemic. [12] As of January 28, 2020, there were more than 5900 confirmed cases and more than 9000 suspected cases of COVID-19 across 33 Chinese provinces or municipalities.[13] Human-to-human transmission of the pathogen was also confirmed.[14] Huang et al. analyzed laboratory-confirmed COVID-19 cases in Wuhan and showed that the symptom onset date of the first patient

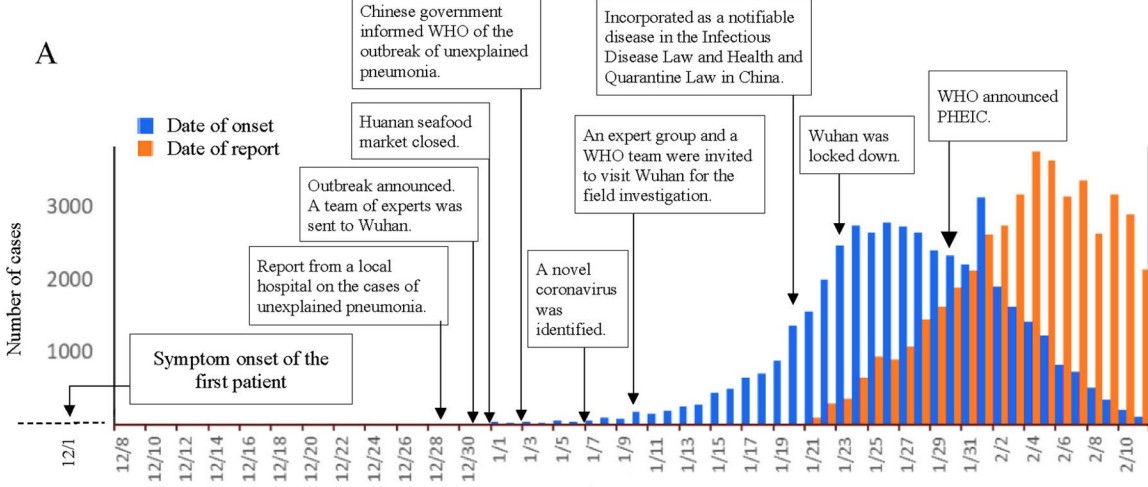

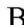

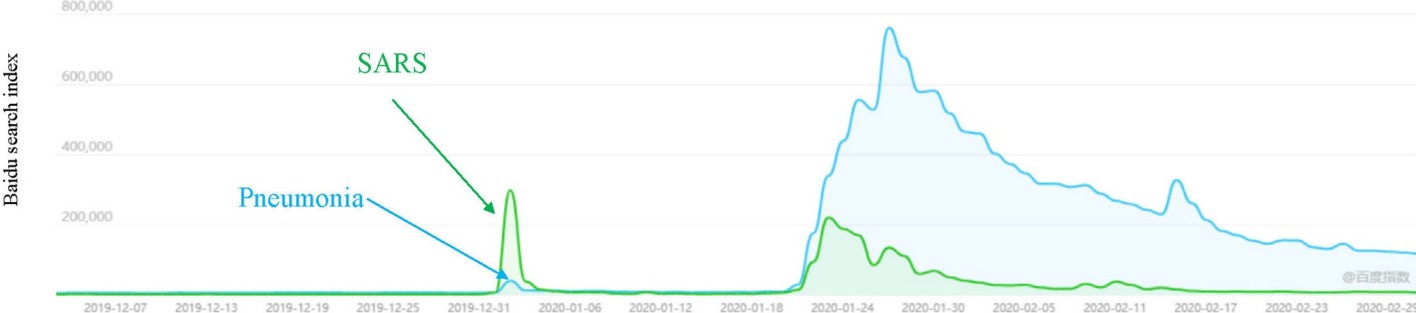

**Fig 1. Comparison of the COVID-19 outbreak and Baidu searching index.** A. Timeline of the COVID-19 outbreak and official response. Only confirmed cases were analyzed referring to the report by The Novel Coronavirus Pneumonia Emergency Response Epidemiology Team in China. [10] B. The online search index for the terms of "pneumonia" and "SARS".

was December 1, 2019.[14] It is estimated that the origin of COVID-19 was most likely earlier than December 2019. As shown in Fig 1A, it took more than one and a half months for the traditional surveillance system to trigger the alert of the outbreak of this EID.

## Signals of the outbreak from the online search index

As shown in Fig 1B, there was a search peak for the terms of "pneumonia" (39641 times) and "SARS" (297864 times) on December 31, 2019, mainly in Wuhan (pneumonia: 11304 times; SARS: 53887 times), where the outbreak of COVID-19 occurred. With the official announcement of the exclusion of SARS and the absence of apparent human to human transmission, the number of searches decreased rapidly the following day. Until around January 20, the BSI of these two terms began to rise again, resulting in a second search peak, which was consistent with the increase in confirmed COVID-19 cases countrywide (Fig 1B).

## Signals of the outbreak based on the influenza surveillance system

Overall, there were differences in the number of ILI in 2014–2019 (F = 8.03, P<0.001). As shown in Fig 2A, the ILI case numbers in 2019 were significantly higher than those reported in

A

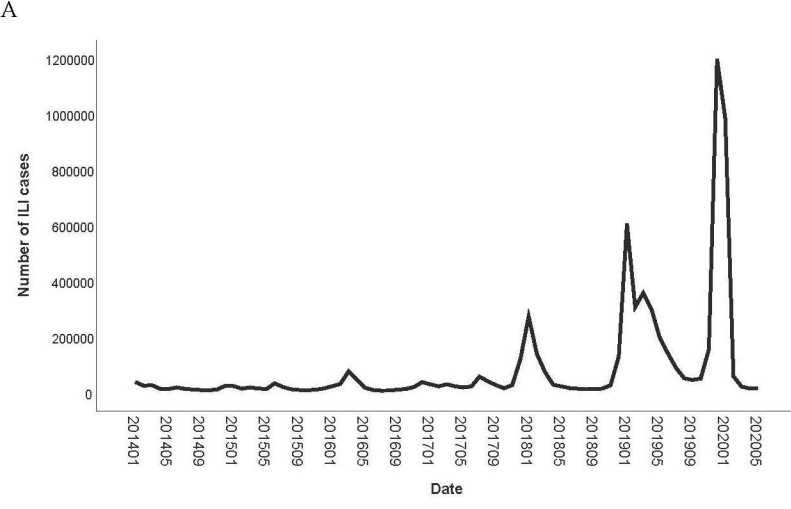

B

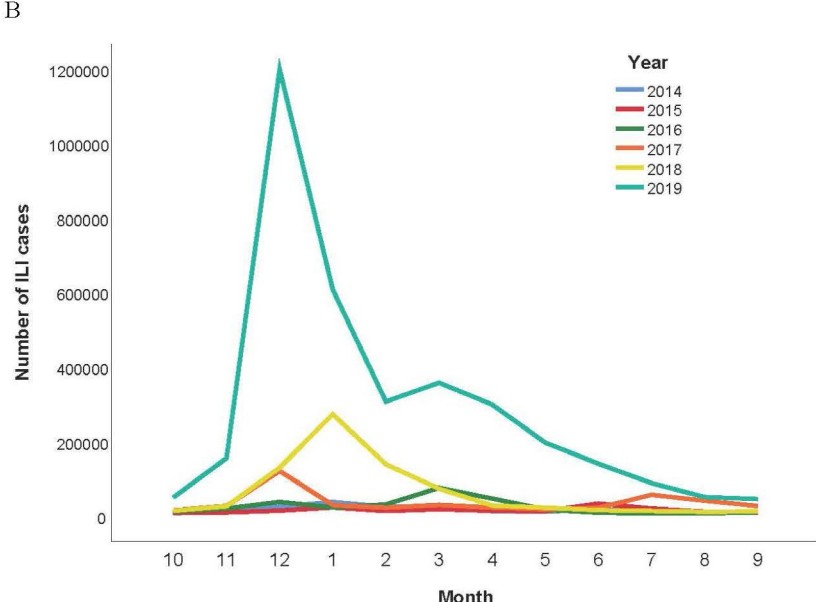

**Fig 2.** Reported influenza-like illness (ILI) cases during 2015–2019 A. The long-term trend of monthly reported ILI cases. B. Comparison of monthly reported ILI cases in different years.

the previous years of 2014–2018 (P<0.05). We observed an early spike in ILI in winter of 2019, with a fast-growing period from November to December (Fig 2B). This observation suggests that COVID-19 cases may have occurred before December 2019. The signal of the abnormally rapid increase in ILI cases was earlier than the report of clinical cases of pneumonia with unknown causes through the official routine disease monitoring system.

## Discussion

Early identification of the emergence of an outbreak of a novel infectious disease is critical to generating a timely response. The traditional monitoring system is adequate for detecting the outbreak of common diseases; however, it is insufficient for the discovery of novel EIDs. In this study, we used COVID-19 as an example to compare the delay time of different tools for

identifying disease outbreaks. The results showed that both the abnormal spike in ILI and the peak of online searches of key terms could provide early signals of novel EIDs.

For centuries, infectious diseases have been among the leading causes of death and have presented growing challenges to human health. The threat is further increased by the continued emergence of new and unrecognized infectious disease epidemics.[15] Due to the lack of sensitive and specific diagnostic tools, infections are often undiagnosed and therefore untreated, or are diagnosed at late stages. Early detection of infectious diseases plays a crucial role in all treatment and prevention strategies.

## The traditional surveillance system has limitations in identifying early signals of the epidemic

A crucial goal of infectious disease surveillance is the early detection of epidemics, which is essential for disease control. In China, the current surveillance system is based on confirmed case reports.[16] It is not practical for health units to perform laboratory tests to confirm a novel infectious disease. Most infectious disease outbreaks start with clinicians noticing unusual patterns. Patients may present with patterns of symptoms that are similar to those of more common diseases but which, after repeated observation and diagnostic testing, may deviate in scale, seasonality, or severity.[17] The discovery of COVID-19 is an example. In December 2019, clinicians from Wuhan City reported several patients with unexplained pneumonia, all of whom were employees of South China seafood wholesale market. Bronchoalveolar lavage samples were collected and sequenced for the whole genome. Bioinformatic analyses indicated that the pathogen was a novel coronavirus, showing the closest relationship with the bat SARS-like coronavirus strain BatCov RaTG13. On January 8, 2020, the novel coronavirus was confirmed as the cause of unexplained pneumonia. However, at that time, people did not realize the potential risk of an epidemic (even less a pandemic) caused by this new pathogen. It was not until mid- to late-January that the risk of widespread transmission was taken seriously. In other words, clinical symptom monitoring and case reporting can help identify new diseases; however, these practices do not provide timely signals of an epidemic.

## Peaks of online searches of key terms provide early signals of an epidemic

Internet-derived information has recently been recognized as a valuable tool for epidemiological investigation.[18] Timeliness and precision in the detection of infectious disease outbreaks from the information published on the web are crucial for prevention of their spread. Arsevska et al. retrieved data from a corpus of relevant documents and compared them with African swine fever (ASF) outbreaks from the Google search engine and the PubMed database.[19] The results showed that relevant documents could serve as a source of terms to detect infectious animal disease emergence on the web. Walker et al. used Google Trends (GT) to investigate whether there was a surge in searches for information related to the COVID-19 epidemic. These authors observed a strong correlation between the frequency of searches for smell-related information and the onset of COVID-19 infection in Italy, Spain, UK, USA, Germany, France, Iran and Netherlands.[20] Li et al. demonstrated that the data obtained from GT, BSI and the Sina Weibo Index on searches for the keywords 'coronavirus and 'pneumonia' correlated with the published daily incidence of COVID-19, with the maximum r > 0.89.[21] However, few studies explored the role of web-based search index in detecting the first occurrence of the COVID-19. In this study, we used the BSI to explore the correlation between the internet search index and the outbreak of COVID-19. The BSI is a public sampling database of search queries users entered into the predominant search engine (Baidu) in China. Unlike GT, the BSI reflects the absolute Baidu search volume and is not displayed as normalized values.[22]

One important issue that emerges from web-based searches is that they tend to underestimate the real epidemiological burden when the general population has poor knowledge of the disease.[18] Additionally, the BSI can be influenced by media clamor. Therefore, the real scientific usefulness of the so-called "digital epidemiology" remains questionable, at least when using GT or BSI. Although the source of information cannot be taken for granted or even replace the "real life" epidemiological data, mining the web is an intriguing perspective for EIDs.

### ILI surveillance and potential of novel EIDs

When and where SARS-CoV-2 originated remains unclear. The similarity between COVID-19 and influenza symptoms makes it possible that the excess ILI cases were due to COVID-19 cases. The presence of SARS-CoV-2-positive swabs in the patients supports this possibility. [23] The predominant symptoms associated with COVID-19 are fever, cough, and sore throat; that is, patients often present with an ILI. At the early stage of the epidemic, COVID-19 cases may have been misdiagnosed as influenza or other respiratory diseases. Thus, we hypothesized that ILI surveillance data could be used as a tool for early detection of COVID-19. Kong et al. analyzed 640 throat swabs collected from patients with ILI in Wuhan from October 6, 2019, to January 21, 2020, and found that nine samples were positive for SARS-CoV-2, suggesting community transmission of SARS-CoV-2 in Wuhan in early January 2020.[24] The dramatic increase in ILI in Wuhan in early December further supported this hypothesis.[24] Spellberg et al. observed a seasonal spike in ILI in Los Angeles, USA. [25] Among patients with mild ILI, 5% were tested positive for SARS-CoV-2. Such transmission is consistent with the countywide unusual third ILI spike that occurred late in the season and with declining rates of influenza positivity.[25] However, seasonal influenza activity was lower in 2020 than in previous years in Japan.[26] It may have been affected by temperature or virulence and by measures taken to constrain the SARS-CoV-2 outbreak.[26] The coinfection of COVID-19 and influenza A reported in Iran also highlighted the importance of considering SARS-CoV-2 PCR assay regardless of positive findings for other pathogens during the epidemic.[27] Silverman et al. explored how ILI outpatient surveillance data could be used to estimate the prevalence of COVID-19, and they found a surge in noninfluenza ILI above the seasonal average in March 2020 and showed that this surge correlated with COVID-19 across states. [28]

In our study, several potential limitations should not be neglected. First, the web-based search for key terms or ILI surge counts in relation to the emergence of COVID-19 may be attributed to potential confounders. Second, the observed ILI surge may represent more than just SARS-CoV-2-infected patients. Whether ILI surveillance data could be used for the signal of the EIDs without dominant features of COVID-19, such as cough and fever, is unclear. Third, the web-based search can be affected by media coverage, the population's knowledge or the degree of information disclosure.

In conclusion, monitoring abnormal surges in ILI and identifying online search peaks of key terms can provide early signals of novel disease outbreaks. We emphasize the importance of testing these findings and discussing the broader potential to use syndromic surveillance, internet searches, and social media data together with traditional disease surveillance systems for early detection and understanding of EIDs.

## Author Contributions

**Conceptualization:** Jianming Wang.

**Data curation:** Yaoyao Dai, Jianming Wang.

**Formal analysis:** Yaoyao Dai, Jianming Wang.

**Funding acquisition:** Jianming Wang.

**Investigation:** Yaoyao Dai, Jianming Wang.

**Methodology:** Yaoyao Dai, Jianming Wang.

**Project administration:** Jianming Wang.

**Resources:** Jianming Wang.

**Software:** Jianming Wang.

**Supervision:** Jianming Wang.

**Validation:** Jianming Wang.

**Visualization:** Yaoyao Dai, Jianming Wang.

**Writing – original draft:** Yaoyao Dai, Jianming Wang.

**Writing – review & editing:** Jianming Wang.

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
