## [Decision Letter · Decision Letter 0]

15 Jul 2020

Dear Prof. Wang,

Thank you very much for submitting your manuscript "Identifying the outbreak signal of COVID-19 before the response of traditional disease monitoring system" for consideration at PLOS Neglected Tropical Diseases. As with all papers reviewed by the journal, your manuscript was reviewed by members of the editorial board and by several independent reviewers. In light of the reviews (below this email), we would like to invite the resubmission of a significantly-revised version that takes into account the reviewers' comments. 

We cannot make any decision about publication until we have seen the revised manuscript and your response to the reviewers' comments. Your revised manuscript is also likely to be sent to reviewers for further evaluation.

Sincerely,

Abdallah M. Samy, PhD

Deputy Editor

Reviewer's Responses to Questions

**Key Review Criteria Required for Acceptance?**

**Methods**

-Are the objectives of the study clearly articulated with a clear testable hypothesis stated?

-Is the study design appropriate to address the stated objectives?

-Is the population clearly described and appropriate for the hypothesis being tested?

-Is the sample size sufficient to ensure adequate power to address the hypothesis being tested?

-Were correct statistical analysis used to support conclusions?

-Are there concerns about ethical or regulatory requirements being met?

Reviewer #1: 1. How can the confounding factors be controlled for "the Baidu search engine (http://index.baidu.com/v2/#/) to analyze Baidu Index for the keywords of “pneumonia” and “SARS” being searched from 1 November, 2019, to 1 February, 2020. " ?

2.Please show the approved documents for the ethical approval.

Reviewer #2: 1. Please confirm that most of the important literature has been included in the paper.

2. The analysis are mainly descriptive, and the authors need to use statistical testing to come to the conclusion if possible.

3. This study is an aggregation analysis, and only need a exemption approval of ethics.

**Results**

-Does the analysis presented match the analysis plan?

-Are the results clearly and completely presented?

-Are the figures (Tables, Images) of sufficient quality for clarity?

Reviewer #1: (No Response)

Reviewer #2: 1. The analysis are reasonable. Is there possbility to analysis the spatial correlation of ILI and COVID-19? The current correlation may not be enough to support the conclusions.

2.The correlation of ILI and COVID-19 may be from a co-occurring event, that is the two events may be occur simultaneously with seasonality, meaning that the next peak of ILI might not be a good signal of COVID-19. 

3. Does the Figure 1A completed by the author my themselves? Otherwise they need to declare the information and cite.

**Conclusions**

-Are the conclusions supported by the data presented?

-Are the limitations of analysis clearly described?

-Do the authors discuss how these data can be helpful to advance our understanding of the topic under study?

-Is public health relevance addressed?

Reviewer #1: How can these abnormal and rapid increase in the number of influenza cases be related to COVID-19, why not other infectious diseases?

Reviewer #2: Conclusion need to be more specific and give detailed information of the measures we can taken to detect the epidemics.

**Editorial and Data Presentation Modifications?**

Reviewer #1: (No Response)

Reviewer #2: (No Response)

**Summary and General Comments**

Reviewer #1: (No Response)

Reviewer #2: This paper performed a comparative study to discuss the feasibility of early warning of the outbreak of COVID-19 in China based on the influenza surveillance data and internet Baidu search index to evaluate the timelines of the alert signals in comparison with the traditional case reporting system and official response, and suggested that monitoring abnormal spike of influenza like cases or identifying online search peak of key terms could provide early signals of novel emerging infectious disease outbreak.

overall this paper provided very useful information about the early detection of COVID-19 epidemics. However, because of the lack of detailed analysis and specific conclusion, the authors need to put more further analysis in the manuscript inorder to have a high impact for the academic prority.

PLOS authors have the option to publish the peer review history of their article (what does this mean?). If published, this will include your full peer review and any attached files.

Reviewer #1: No

Reviewer #2: No
---

## [Decision Letter · Decision Letter 1]

28 Aug 2020

Dear Prof. Wang,

We are pleased to inform you that your manuscript 'Identifying the outbreak signal of COVID-19 before the response of the traditional disease monitoring system' has been provisionally accepted for publication in PLOS Neglected Tropical Diseases.

Best regards,

Abdallah M. Samy, PhD

Deputy Editor
